# Fatty Acyl Esters of Hydroxy Fatty Acid (FAHFA) Lipid Families

**DOI:** 10.3390/metabo10120512

**Published:** 2020-12-17

**Authors:** Paul L. Wood

**Affiliations:** Metabolomics Unit, College of Veterinary Medicine, Lincoln Memorial University, 6965 Cumberland Gap Pkwy, Harrogate, TN 37752, USA; paul.wood@lmunet.edu

**Keywords:** FAHFA, OAHFA, ornithine, wax esters, ω-hydroxylation

## Abstract

Fatty Acyl esters of Hydroxy Fatty Acids (FAHFA) encompass three different lipid families which have incorrectly been classified as wax esters. These families include (i) Branched-chain FAHFAs, involved in the regulation of glucose metabolism and inflammation, with acylation of an internal branched-chain hydroxy-palmitic or -stearic acid; (ii) ω-FAHFAs, which function as biosurfactants in a number of biofluids, are formed via acylation of the ω-hydroxyl group of very-long-chain fatty acids (these lipids have also been designated as o-acyl hydroxy fatty acids; OAHFA); and (iii) Ornithine-FAHFAs are bacterial lipids formed by the acylation of short-chain 3-hydroxy fatty acids and the addition of ornithine to the free carboxy group of the hydroxy fatty acid. The differences in biosynthetic pathways and cellular functions of these lipid families will be reviewed and compared to wax esters, which are formed by the acylation of a fatty alcohol, not a hydroxy fatty acid. In summary, FAHFA lipid families are both unique and complex in their biosynthesis and their biological actions. We have only evaluated the tip of the iceberg and much more exciting research is required to understand these lipids in health and disease.

## 1. Introduction

In this review, we present an abbreviated overview of three major Fatty Acyl esters of Hydroxy Fatty Acids (FAHFA) lipid families which individually serve unique metabolic and/or structural functions (Figure 1). Branched-chain FAHFAs are essential modulators of metabolic and inflammatory health status. These FAHFAs are formed by the acylation of medium chain (16 to 18 carbons) hydroxy fatty acids.

ω-FAHFAs are unique in that they are synthesized by the acylation of very-long-chain ω-hydroxy fatty acids. This synthetic pathway requires complex metabolic machinery for the biosynthesis of very-long-chain ω-hydroxy fatty acids (30 to 34 carbons). ω-FAHFAs, which have also been termed o-acyl hydroxy fatty acids (OAHFA), are biosurfactants in the tear film of the eye, amniotic fluid, semen, sperm, skin, and the vernix caseosa.

Ornithine-FAHFA are bacterial lipids synthesized by the acylation of short-chain 3-hydroxy fatty acids and the addition of ornithine to the free carboxy group of the hydroxy fatty acid. These complex acylated ornithines act as a permeability barrier in the membranes of bacteria.

## 2. FAHFA Lipid Families

### 2.1. Branched-Chain FAHFA

The discovery of branched-chain FAHFAs (Figure 1) was based on their efficacy as endogenous anti-diabetic and anti-inflammatory lipids [1]. The anti-diabetic actions appear to involve selective agonism at G-protein-coupled receptor40 (GPR40), thereby augmenting glucose stimulation of insulin release [2]. The internal branched FAHFAs are a diverse lipid class of at least 51 FAHFA families and 301 regioisomers [3]. Hence, high-resolution chromatography coupled with mass spectrometry is essential to assess the potential roles of different families in physiology [3,4,5].

The most studied FAHFAs are the palmitic-hydroxystearic (PAHSA) and palmitic-hydroxypalmitic (PAHPA) families, which include the 5-, 7-, 8-, 9-, 10-, 11-, 12-, and 13-hydroxy regioiomers [5,6]. While these FAHFAs undergo de novo synthesis in most organs, they also are present in dietary plant sources [7,8]. Biosynthesis of branched-chain FAHFAs can be augmented by oxidative stress [9] and involves the transfer of a fatty acid from a fatty acid-CoA to a hydroxy fatty acid via lipid acyltransferases [10,11]; see Figure 2. The human gene candidates for this acylation are ACNATP (acyl-CoA:amino acid *N*-acyltransferase) and BAAT (bile acid CoA: amino acid *N*-acyltransferase).

The metabolism of FAHFAs involves transmembrane threonine and serine hydrolases. These include androgen induced gene 1 (AIG1), androgen-dependent TFPI-regulating protein (ADTRAPP), and carboxyl ester lipase (CEL; MODY8), which hydrolyze a FAHFA to the free fatty acid and hydroxy fatty acid [12,13,14]. 

Recent data have also demonstrated that branched-chain FAHFAs can further acylate the free hydroxy group at sn2 of diacylglycerols and that this represents a large neutral lipid storage pool of branched-chain FAHFAs in tissues, with lipolytic enzymes releasing the free FAHFA [15].

MS^2^ characterization of branched-chain FAHFAs (Table 1 and Table 2) provides the structural information regarding the fatty acid and hydroxy fatty acid substituents but does not define the position of the hydroxy group in the hydroxy fatty acid chain [1]. Hence, high-resolution chromatography, prior to mass spectrometric analysis, is required to quantitate the individual isobars [4,5].

### 2.2. ω-FAHFA

ω-FAHFAs (Figure 1) differ from branched-chain FAHFAs in that the hydroxy fatty acid is a very-long-chain fatty acid (VLCFA) with a terminal (ω) hydroxyl group, formed via cytochrome P-450 (CYP450) ω-hydroxylases [16]. This contrasts with the 16- to 18-carbon hydroxy fatty acids of branched-chain FAHFAs. ω-FAHFAs, which have also been termed o-acyl-ω-hydroxy fatty acids (OAHFA), are restricted to meibomian glands and the associated tear film of the eye [17,18,19,20,21], amniotic fluid [22], sperm [23], semen [23], skin [24,25], and the vernix caseosa covering the skin of newborns [26]. The vernix caseosa also contains large amounts of cholesterol esters of ω-FAHFAs [26]. 

ω-FAHFAs are amphiphilic lipids that act as biosurfactants in the biofluids listed above. The diversity of these lipids is presented in Table 1. The functions of ω-FAHFAs are conjectured to involve roles as biosurfactants in the tear film [17] and amniotic fluid [22]. In the case of amniotic fluid, the levels of these lipids dramatically spike at 9 months of fetal development, suggesting a role in the delivery process [22] as a component of the vernix caseosa [26].

The unusual feature of ω-FAHFAs is the ω-hydroxy-VLCFA component, which is 30 to 34 carbons long. Hence, ELOngation of Very Long chain fatty acids (ELOVLs) are critical enzymes involved in the elongation fatty acids in the biosynthesis (Table 1) of ω-FAHFAs [18,27,28,29]. CYP4F, CYP4X, and CYP51A are next involved in the formation of the ω-hydroxy group [16,29]. The lipid acyltransferases involved in the final step of ω-FAHFA biosynthesis remain to be characterized.

MS^2^ characterization of ω-FAHFAs (Table 2) provides structural information regarding the fatty acid and hydroxy VLCFA substituents [22,23].

### 2.3. Ornithine-3-FAHHA (ORN-FAHFA)

Bacterial ORN-FAHFA biosynthesis [30,31,32]—see Figure 2—involves the N-acylation of ornithine with a 3-hydroxy fatty acid by ORN-Nα-acylase B (OlsB; EC 2.3.2.30). Next, ORN-FAHFA is generated via o-acylation of the hydroxy fatty acid by ORN lipid O-acyltransferase A (OlsA; EC 2.3.1.270). Environmental stress can induce the synthesis of ORN-FAHFAs [30] and can also induce sequential methylation (Me) of the arginine nitrogen to generate the methyl, dimethyl, and trimethyl derivatives [33].

ORN-FAHFAs are present in 25% of bacterial species and are predominantly localized to the outer membrane of Gram-negative bacteria [30,31], where they stabilize the membrane and act as a permeability barrier. It has been postulated that the zwitterionic nature of ORN-FAHAs allows them to stabilize the negative charge of bacterial membrane lipopolysaccharide, thereby increasing the hydrogen bonding between membrane lipid molecules and decreasing membrane permeability [30].

MS^2^ characterization of ORN-FAHFAs [32], Me-Orn-FAHFAs [33], Di-Me-Orn-FAHFAs [33], and Tri-Me-Orn-FAHFAs [33] provides structural information regarding the fatty acid and hydroxy fatty acid substituents (Table 2).

### 2.4. Wax Esters

In contrast to FAHFA lipid families, which are formed by the acylation of a hydroxy fatty acid, wax esters (WE) are generated by the acylation of a fatty alcohol [34,35,36,44]; see Figure 2. First, fatty-acyl-CoA reductase (FAR; EC 1.2.1.84) converts a fatty acyl-CoA to the corresponding fatty aldehyde, which in turn is converted to the fatty alcohol by fatty aldehyde reductase (FALR; EC 1.1.1.2). Acylation of the fatty alcohol involves acyl-CoA wax alcohol acyltransferase (AWAT1; EC 2.3.1.75). WE metabolism occurs via wax ester hydrolase (WEH; EC 3.1.1.50).

In mammals, wax esters are produced by several exocrine glands and are actively secreted. These include ocular meibum produced by meibomian glands [37]; sebum produced by sebaceous glands of hair follicles [38,39,40] which is a component of the epidermis [41] and hair [42]; and saliva, produced by the salivary glands [43]. Wax esters as components of the vernix caseosa [25,26] probably act as lubricants during delivery. Wax esters clearly play physiological roles as lubricants as a result of their neutral charge, contrasting with the polar nature of FAHFAs.

In bacteria, wax esters function as storage pools for energy and carbon, with these lipids representing the major storage pool and acting as an energy source for bacteria [45].

## 3. Conclusions

FAHFA lipid families are both unique and complex in their biosynthesis and their biological actions. We have only evaluated the tip of the iceberg and much more exciting research is required to understand these lipids in health and disease. This research will involve the use of transgenic models, characterization and cloning of biosynthetic enzymes, and high-resolution mass spectrometric lipidomics analyses of human disease populations. Such research may result in new therapeutic approaches for diabetes and inflammatory diseases in the case of branched-chain FAHFAs. With ω-FAHFAs, this research has the potential to lead to new approaches to fertility and newborn delivery. Understanding ORN-FAHFAs ultimately may lead to new antimicrobial tactics.

## Figures and Tables

**Figure 1 metabolites-10-00512-f001:**
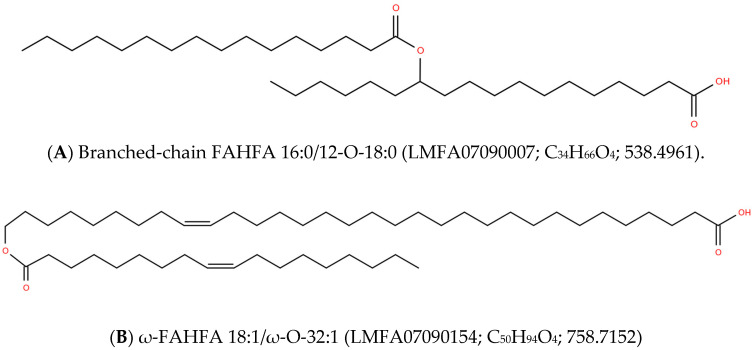
Structures of fatty acyl esters of hydroxy fatty acids (FAHFA) and wax esters (WE). The acyl linkages are formed by the following reaction: R-OH + R′-COOH = R-O-CO-R′ +H_2_O. In eukaryotes, the major branched-chain FAHFAs (**A**) are palmitic acid/hydroxystearic acid (16:0/O-18:0) and oleic acid/hydroxystearic acid (18:1/O-18:0), with the hydroxyl group at carbons 5, 9, or 12 of stearic acid. The ω-FAHFAs, represented in (**B**), are unique in that the hydroxy fatty acids are long-chain fatty acids with a terminal (ω) hydroxyl group. In the case of ornithine (ORN)-FAHFA (**C**), the hydroxy fatty acid is consistently a 3-hydroxy fatty acid and the addition of ORN is to the free carboxyl group of the hydroxy fatty acid component of the FAHFA. In contrast, wax esters (WE; **D**) are formed by the acylation of a fatty alcohol, resulting in the formation of a neutral lipid, which contrasts with the charged FAHFA lipids. While Lipid Maps categorizes FAHFAs as wax esters, the completely different biosynthetic pathways (i.e., fatty acid vs. fatty alcohol addition) indicates that this is too simplified and incorrect. The Lipid Maps number and exact mass of the lipids are also provided.

**Figure 2 metabolites-10-00512-f002:**
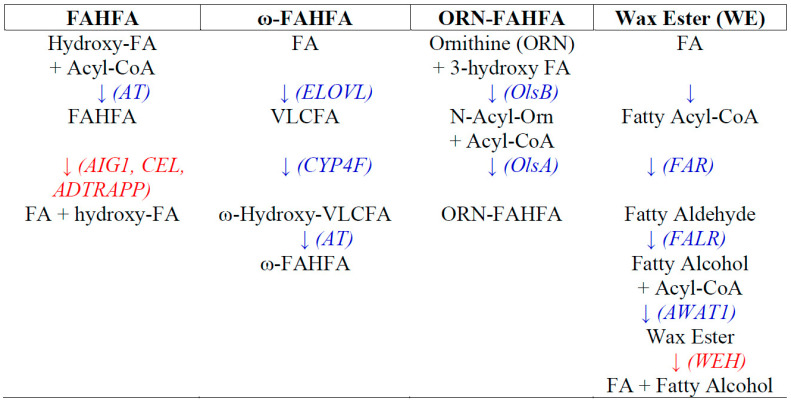
Biosynthetic pathways and metabolism of fatty acyl esters of hydroxy fatty acids (FAHFA), ω-FAHFA, ornithine-FAHFA (ORN-FAHFA), and wax esters (WE). FAHFA: *AT*, unidentified acyltransferase; *AIG1*, androgen induced gene 1; *ADTRAPP*, androgen-dependent TFPI-regulating protein; *CEL*, carboxyl ester lipase; FA, fatty acid. ω-FAHFA: *ELOV*L, acetyl-CoA-acyltransferase; *CYP4F*, cytochrome P450 monooxygenase (EC 1.14.14.1); *AT*, unidentified acyltransferase. VLCFA, very-long-chain fatty acid. ORN-FAHFA: *OlsB*, L-ORN-N-α-acylase (EC 2.3.2.30); *OlsA*, ORN lipid o-acyltransferase (EC 2.3.1.270). WE: *FAR*, fatty-acyl-CoA reductase (EC 1.2.1.84); *FALR*, fatty aldehyde reductase (EC 1.1.1.2); *AWAT1*, acyl-CoA wax alcohol acyltransferase (EC 2.3.1.75); *WEH*, wax ester hydrolase (EC 3.1.1.50).

**Table 1 metabolites-10-00512-t001:** Molecular diversity of ω-FAHFAs in equine sperm [23] and amniotic fluid [22]. Note that the ω-fatty acids are composed of 30 to 34 carbons, contrasting with the branched-chain FAHFAs in which the hydroxy fatty acids are 16 to 18 carbons (e.g., PAHSA 16:0/18:0(OH)). In these biological specimens, these were the easily detectable (i.e., dominant) ω-FAHFAs.

ω-FAHFA	Fatty Acid	ω-Hydroxy Fatty Acid
46:1	14:0	32:1
	16:0	30:1
46:2	16:1	30:1
	14:1	32:1
48:1	16:0	32:1
	18:1	30:0
48:2	16:1	32:1
	18:1	30:1
50:1	16:0	34:1
	18:0	32:1
50:2	18:1	32:1
	16:1	34:1
50:3	18:2	32:1
51:2	18:1	33:1
52:2	18:1	34:1
52:3	18:2	34:1
	18:0	34:3

**Table 2 metabolites-10-00512-t002:** Summary of the occurrence, functions, and MS^2^ characterization of FAHFA lipid families.

Lipid Class	Occurrence	Functions	MS^2^
Branched-Chain FAHFA [3,4,5,6,7,8,9]	Tissues Blood Plants	Glucose regulation Anti-inflammatory	[M − H]^−^ ↓ [FA]^−^ > [Hydroxy-FA]^−^
Branched-Chain FAHFA—DAG [15]	Adipose tissue	Branched-chain FAHFA storage pool	[M + NH_4_]^+^ ↓ [DAG—H_2_O]^+^ [M-FA-H_2_O]^+^ [M-Hydroxy-FA-H_2_O]^+^
ω-FAHFA [17,18,19,20,21,22,23,24,25]	Meibium Tear film Amniotic fluid Sperm and semen Skin Vernix caseosa	Biosurfactants	[M − H]^−^ ↓ [FA]^−^ > [Hydroxy-FA]^−^
ω-FAHFA—CE [26]	Vernix caseosa	ω-FAHFA storage pool	[M + H]^+^ ↓ [M-(cholesterol-H_2_O)]^+^
ORN-FAHFA [30,31,32]	Bacteria	Membrane stabilization Permeability barrier	[M + H]^+^ → [M-(FA-H_2_O)]^+^ [M − H]^−^ → [M-FA]^−^
Methyl-ORN-FAHFA [33]	Bacteria	Membrane stabilization in acidic environment	[M + H]^+^ → [M-FA]^+^ > [M-FA-CH_5_N]^+^
Tri-Methyl-ORN-FAHFA [33]	Bacteria	Membrane stabilization in acidic environment	[M + H]^+^ → [M-FA]^+^ > [M-FA-C_2_H_7_N]^+^
Tri-Methyl-ORN-FAHFA [33]	Bacteria	Membrane stabilization in acidic environment	[M + H]^+^ → [M-FA]^+^ > [M-FA-C_3_H_9_N]^+^
Wax Ester [34,35,36,37,38,39,40,41,42,43]	Meibium Tear film Sebum Epidermis Saliva Vernix caseosa Hair Plants Bacteria	Barrier to moisture loss Lubricant	[M + NH_4_]^+^ ↓ [FA + H]^+^

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
