# Peer review of "Fatty Acyl Esters of Hydroxy Fatty Acid (FAHFA) Lipid Families"

_metabolites, 2020, doi:10.3390/metabo10120512_

Round 1
Reviewer 1 Report
The review "Fatty Acyl Esters of Hydroxy Fatty Acid (FAHFA) Lipid Families" is well written and easy to follow. The information presented therein is practical and useful to the scientific community, and the readership. The reviewer did not find any major issues with the manuscript.
Few minor comments:
Some of the acronyms and abbreviations should be defined when they first appear in the text.
Please consider adding references in Table 2 (columns 2, 3 and 4).
Also in Table 2, the accurate mass of the fragment is not needed (six decimal points are too many even for the most accurate mass spec).
The abstract and the manuscript both miss a conclusion and possible directions of the research involving the FAFHAs.
Author Response
File attached.

Reviewer 2 Report
Review
Paul L. Wood
Fatty Acyl Esters of Hydroxy Fatty Acid (FAHFA) Lipid Families
A comprehensive but brief review on several classes of FAHFA’s is written nicely in an easy to read manuscript by Wood.
The review is timely, to my knowledge there are no other reviews on this specific topic including the different FAHFA with a comparison to wax esters. Other literature seems to focus more on the branched FAHFA since they are more clearly involved human biological activities with respect to disease and possible treatment of including obesity and diabetes type II, such as the recent review by Brechjova et. al. https://doi.org/10.1016/j.plipres.2020.101053
However, the review by Wood has clearly divided the focus over the three FAHFA families and compares to wax esters.
It is a great overview of what is known about FAHFA’s, their (MS) analysis, and their known roles. What is missing in my opinion is a bit more the outlook. Which questions can be answered in future research on FAHFA’s? What is needed for this?
When elaborated a bit more on which knowledge is lacking so far on FAHFA and how this can be investigated in the future it would become a very useful review for many readers to learn about FAHFA, which would suit well to be published in metabolites in my opinion. Then the review would be a great starting point for student and scholars wanting to learn more about FAHFA’s, and interest in these lipids is clearly on the rise.
Comments.
Title. No dot needed in title.
Abstract. Abstract is to the point and covers the main points of the main text.
Line 2. “incorrectly been classified as was esters” This is not something I could find again in the main text.
1. Introduction
page 1. Line 1. Abbreviation FAFHA is explained in abstract, however, as far as I am aware the first time in the main text the abbreviation has also to be written completely spelled out.
Page 1. Line 4. should end with a dot “.”
page 2. Figure 1. It would be nice if the different structures were in the same size, drawn with the same program. The bonds have different sizes now, the WE molecule is clearly in different a scale then the branched-chain FAHFA for instance.
Page 2 Figure 1 serves the purpose very well. An addition could be to show the reaction; structure of a fatty acid and structure of a hydroxy fatty acid →structure of FAHFA + (water)
I noticed this is depicted in fig 2 but only with the written chemical names. But I think the target-audience of this review could really appreciate this with structures early on in the review. This will help to understand where the new bond is formed. So please it include in fig. 1.
page 2. Fig 1. It is nice that the Lipid Maps Identifiers are included in the figure, however for A and for B the LMID did not yield any result on the lipid maps website. C and D are correct and can be found on the LM database. (visited the lipid maps website https://lipidmaps.org/data/structure/index.php on November 18, 2020). Please check if the numbers are correct.
Page 2. Fig 1. In fig 1. a major difference between WE and FAHFA is described, namely the fact that WE are neutral lipids. I would suggest to also mention this in the main text, in the part on WE.
2. FAHFA Lipid Families
Page 2. subtitle 2.1. Why is Chain written with a capital in “Branched-Chain FAHFA”?
Page 3. line 6. “Hence, high resolution chromatography … families in physiology[3-5].” A recent review in Biomolecules on analytical methods for FAHFAs might be useful to include in the references. Readers interested in analytical methods can find a collection of methods there, from sample preparation to instrumentation that is used in FAHFA analysis.
Maroula G. Kokotou, Analytical Methods for the Determination of Fatty Acid Esters of Hydroxy Fatty Acids (FAHFAs) in Biological Samples, Plants and Foods Biomolecules 2020, 10, 1092; doi:10.3390/biom10081092
page 3. Fig. 2. Captions usually appear below figure.
Page 4. line 5. “Recent data has … the free FAHFA [15].” Please specify this becomes a special class of triacylglycerol molecule. What is also noteworthy to mention is that it becomes a neutral lipid, this is respect to why it can become a large storage pool.
Page 4. line 29. “The lipid acyltransferases … to be characterized.” Could you elaborate on this search for enzymes? Which steps are taken already or have to be taken?
page 5. Table 1. Why do certain combination not exist? For instance only 1 combination for fatty acid 14:0. Or are these the only ones experimentally detected? Please clarify in text or legend of table 1.
page 5. line 6. “ORN-FAHFAs are present … a permeability barrier.” Please elaborate on how this type FAHFA helps to stabilize the membrane. By which mechanism this occurs? Also, please elaborate on why this helps in increase the barries function of the membrane.
Could these functions be performed by the other FAHFA lipid families? In other words, does the structure of a FAHFA lipid family define its function? This is something that could also be elaborated on in later on in the review for all three FAHFA families mentioned in this review.
Page 5. line 13. 2.4 Wax Esters
Only the function in mammals is mentioned, bacteria and most likely other organisms also have wax esters with certain functions. Since this review is not only focused on mammals, otherwise the Bacterial ORN-FAHFA would not have been mentioned, it is good to also include some bacterial wax esters and their roles in bacteria.
Page 6. Conclusion. “FAHFA lipid families … health and disease.” I agree, but as a reader I would be very happy to read a bit more on which “exciting research is required” and what we possible could expect from this. Please elaborate.
Author Response
File attached.

Round 2
Reviewer 2 Report
all points are addressed adequately. Except one little detail misses explanation or correction and then it is OK for being published;
Page 3. line 6. “Hence, high resolution chromatography … families in physiology[3-5].” A recent review in Biomolecules on analytical methods for FAHFAs might be useful to include in the references. Readers interested in analytical methods can find a collection of methods there, from sample preparation to instrumentation that is used in FAHFA analysis.
Maroula G. Kokotou, Analytical Methods for the Determination of Fatty Acid Esters of Hydroxy Fatty Acids (FAHFAs) in Biological Samples, Plants and Foods Biomolecules 2020, 10, 1092; doi:10.3390/biom10081092
Reference added.
Good to see the biomolecules paper was included. But also the Nature Protocol reference was removed. This was not necessary in my opinion. If on purpose explain why, if accidental correct it please. thanks.
Author Response
The suggested reference was a review of methods and replaced a focused method. Also there already are several other methods references such that the one replaced is not essential.